# Research on SPAD Estimation Model for Spring Wheat Booting Stage Based on Hyperspectral Analysis

**DOI:** 10.3390/s24051693

**Published:** 2024-03-06

**Authors:** Hongwei Cui, Haolei Zhang, Hao Ma, Jiangtao Ji

**Affiliations:** 1College of Agricultural Equipment Engineering, Henan University of Science and Technology, Luoyang 471000, China; 2Longmen Laboratory, Luoyang 471000, China

**Keywords:** spring wheat, booting stage, SPAD, fractional-order differentiation, SMA-LSSSVM, fractional-order differentiation spectral index

## Abstract

With the rapid progression of agricultural informatization technology, the methodologies of crop monitoring based on spectral technology are constantly upgraded. In order to carry out the efficient, precise and nondestructive detection of relative chlorophyll (SPAD) during the booting stage, we acquired hyperspectral reflectance data about spring wheat vertical distribution and adopted the fractional-order differential to transform the raw spectral data. After that, based on correlation analysis, fractional differential spectra and fractional differential spectral indices with strong correlation with SPAD were screened and fused. Then, the least-squares support vector machine (LSSSVM) and the least-squares support vector machine (SMA-LSSSVM) optimized on the slime mold algorithm were applied to construct the estimation models of SPAD, and the model accuracy was assessed to screen the optimal estimation models. The results showed that the 0.4 order fractional-order differential spectra had the highest correlation with SPAD, which was 9.3% higher than the maximum correlation coefficient of the original spectra; the constructed two-band differential spectral indices were more sensitive to SPAD than the single differential spectra, in which the correlation reached the highest level of 0.724. The SMA-LSSSVM model constructed based on the two-band fractional-order differential spectral indices was better than the single differential spectra and the integration of both, which realized the assessment of wheat SPAD.

## 1. Introduction

Wheat as a staple crop is of paramount importance in the world, and its growth and development status is closely related to yield quality, so accurate and effective monitoring of wheat growth is crucial for improving yield and ensuring food security [1]. A large amount of studies have been carried out to monitor the growth status of wheat by measuring its phenotypic parameters, among which chlorophyll content is an important characterization parameter reflecting the growth status of the crop, which is related to the chlorophyll fluorescence reaction of the crop leaves [2] and is the basis for the exchange of substances and energy between the crop and the external environment, and it can mirror the growth status and the degree of health of the crop.

Traditional wheat SPAD monitoring mainly involves field surveys and sampling methods, which have the disadvantages of being destructive, inefficient and costly [3,4,5]. With the development of hyperspectral technology, hyperspectral data, compared with traditional multispectral data, provide a data basis for the fine detection of targets due to the advantages of larger resolution and narrower wavelength bands, while differentiation of reflectance spectral data can enhance the subtle changes of gradient in spectral profiles and avoids the signal loss of integer-order differentiation prone to eliminating noise. Therefore, the fractional-order differentiation, as an extension of integer-order differentiation, can highlight the subtle information of spectra, describe the small differences between spectral data, enhance the absorbance properties of weak spectra to a point and retain more valid information [6,7,8,9].

In recent years, a number of domestic and international studies have attempted to improve the valuation accuracy of crop phenotypic detection by applying fractional-order differentiation, involving aspects such as pests and diseases [10], chlorophyll [11] and soil organic matter [12,13,14,15,16,17]. Currently, spectral data processing for wheat mainly includes the first-order derivative transform, logarithmic transform and even wavelet energy coefficient method, but most of them focus on the excavation of spectral data themselves, failing to adequately combine and retain more detailed information to enhance the ability of spectral data to detect plant growth changes. Ding et al. [18] extracted features from raw spectral data by the continuous wavelet transform, spectral differentiation and vegetation index method, in which the most sensitive spectral differentiation feature for winter wheat fusarium head blight was under first-order differentiation, and its correlation coefficient was 0.58. Chen et al. [19] processed spectral data using fractional-order differentiation (1st–2nd order) at an interval of 0.1 to screen the sensitive bands, and the results showed that the hyperspectral data processed by 1.1 order differentiation could provide an inverse prediction of the total nitrogen content of the soil using a BP neural network model in the study area. Li et al. [20] utilized fractional-order differentiation with winter wheat canopy hyperspectral reflectance data, screened out the spectra that were more sensitive to changes of LAI and used optimal subset regression and a support vector mechanism to build LAI evaluation models for different growth periods. The correlation coefficients between the red-edge peak area and apple nitrogen content obtained by fractional differentiation and logarithmic transformation treatments by Peng [21] et al. reached 0.6 and above. The results of the above studies show the feasibility of using fractional-order differentiation for SPAD monitoring in wheat, but the above studies were mainly carried out to refine the original spectral data. The effectiveness of two-band differential spectral indices [22], fused differential spectral indices and single differential spectra for monitoring wheat phenotypes is unknown which is not conducive to the robustness of wheat-monitoring models.

In summary, few studies have focused on wheat SPAD detection based on fractional-order differentiation. Therefore, spring wheat was investigated to explore the possibility of fractional-order differential spectroscopy to develop SPAD prediction models of wheat at the booting stage in this study. It mainly aims: (1) to analyze the correlation of single spectral and two-band spectral indices with wheat SPAD using fractional-order differential; (2) to analyze the potential of single and two-band differential spectral indices and the fusion of single and two-band differential spectral indices for detecting wheat SPAD and determine the model optimization used for the SPAD prediction.

## 2. Data and Methods

### 2.1. Experimental Material

The booting stage of wheat is a crucial period for increasing grain number and weight throughout the entire growth period. With the rebound of temperature, the fertilizer and water requirements of wheat are relatively high during this period. Therefore, to ensure the final yield of wheat, management of the booting stage is important. The SPAD indirectly reflects the nutritional demand of wheat, and wheat leaves are the main storage and supply source of nutrients. Therefore, the SPAD in wheat leaves is one of the most important indicators for calculating the nutritional composition and yield of wheat. In this study, a spring-wheat-potting experiment was conducted at the Henan University of Science and Technology Experimental Station (Figure 1) in 2023. Before sowing, 3.89 g of urea and 1.29 g of potassium dihydrogen phosphate were applied to each pot in reference to Liang’s potting experiment [23]. The spring wheat varieties used in the experiment were Jinqiang 9, Jinqiang 11 and Longchun 23, and four levels of nitrogen fertilizer urea were applied: 0 kg/hm^2^, 150 kg/hm^2^, 180 kg/hm^2^ and 210 kg/hm^2^ with four replications. The key cultivation techniques of Jinqiang 11 were timely irrigation during the tillering stage and topdressing with urea of 150 kg·hm^−2^. During the jointing period, timely fertilization and irrigation can be applied, and 225.0 kg· hm^−2^ of urea can be topically applied. Therefore, we have set a nitrogen fertilizer application gradient to prevent seedling burning. The experiment adopted artificial control of soil moisture content from sowing to maturity to maintain the soil moisture content at an optimum level for wheat of 60–80%. The soil moisture was monitored using a portable soil detector (SN-3000-*-USB, Prsens, Jinan, China). When the soil moisture content was detected to be below 60%, the soil moisture content was adjusted by watering appropriately to maintain it at 60–80%. Fifteen seeds were sown in each pot uniformly on 13 March 2023. The experimental site is shown in Figure 2.

### 2.2. Data Acquisition

In this study, reflectance data of wheat under different gradients of nitrogen stress were measured using an Ocean Insight spectrometer (Ocean Optics Asia, Florida, USA) on 24 April 2023, and all measurements were made in the laboratory. The Ocean Optics USB4000 spectrometer detector is responsive from 200 to 1100 nm, and the interval of the band is 0.22 nm. A wheat plant in a pot was randomly selected for observation each time, and the fiber-optic probe was about 1 cm away from the wheat leaves during the measurement, which was divided into three layers in the vertical direction from top to bottom. The reflectance data were corrected with a white board before and after each measurement. The general architecture of the wheat-phenotyping system is shown in Figure 3. After the device was connected according to Figure 3, the computer and light source were powered on. The light probe is firstly calibrated on a white board with a reflectivity of 100%, and then inserted into a black box near the top of the wheat leaves to determine the hyperspectral data.

The chlorophyll content detector (SPAD-502, Tokyo, Japan, Konica Minolta) was used to observe the vertical distribution of wheat. When the experiment entered the booting stage, the wheat structure was characterized and plant height was stable and the experimental conditions were available to measure the SPAD data of vertical distribution. At this time, the growth of wheat enters the strong demand period for water and fertilizer, the relative chlorophyll content can visually express the growth status of wheat, and the SPAD of different parts can consequently reflect the nutrient transfer during the growth phase of wheat, so it is indispensable to survey the vertical distribution data of its SPAD. The SPAD was observed in 3 layers in the vertical direction. The first layer was located in 1/2 of the top leaf (SPAD_CL_), the second layer was located in 1/2 of the spike leaf (SPAD_PL_) and the third layer was in 1/2 of the leaf close to the ground (SPAD_RL_). Each layer was measured three times and the mean was recorded as the observation.

### 2.3. Fractional-Order Differentiation

Fractional-order differentiation is an extension of integer-order differentiation. Although it has similar properties to integer-order differentiation, the fractional-order differentiation is more explanatory about the subtle changes and holistic information of spectra. There are three common types of fractional-order differentiation: Grünwald–Letnikov, Riemann–Liouville and Caputo. The fractional-order differentiation used stems from the Grünwald–Letnikov difference of unitary function [24], which transforms the hyperspectral reflectance data with 0–1 order at an interval of 0.1 under fractional-order differentiation, and the *α*-order differentiation formula is:(1)$$\frac{d^{\alpha }f\left(x\right)}{dx^{\alpha }}\approx f\left(x\right)+\left(-\alpha \right)f\left(x-1\right)+\frac{\left(-\alpha \right)\left(-\alpha +1\right)}{2}f\left(x-2\right)+\cdots +\frac{\Gamma \left(-\alpha +1\right)}{n!\Gamma \left(-\alpha +n+1\right)}f\left(x-n\right)$$
where *x* is a correspondence point, *α* is the fractional-order differentiation, Γ is the gamma function and *n* is the difference between the upper and lower differential limits; when *n* is 0, the formula indicates the original data.

### 2.4. Fractional-Order Differential Spectral Index

In this paper, the original spectral indices is referenced in order to reimagine the optimal spectral index. The three kinds of two-band fractional-order differential spectral indices are constructed: fractional-order differential difference index (FDI), fractional-order differential ratio index (FRI), fractional-order differential normalized difference index (FNDI) [25].
(2)$$\mathrm{FDI}={R_{i}}^{\alpha }-{R_{j}}^{\alpha }$$
(3)$$\mathrm{FRI}=\frac{{R_{i}}^{\alpha }}{{R_{j}}^{\alpha }}$$
(4)$$\mathrm{FNDI}=\frac{{R_{i}}^{\alpha }-{R_{j}}^{\alpha }}{{R_{i}}^{\alpha }+{R_{j}}^{\alpha }}$$
where *R* is the spectral reflectance, *α* is the spectral order, meanwhile wavelength *i* is not equal to *j*.

### 2.5. Slime Mold Alrorithm [26] (SMA)

SMA is a heuristic optimization method that grew out of the natural vibration model of slime molds. SMA is based on adaptive weights that allow the method to avoid falling into a local optimum while maintaining a fast convergence rate.

SMA utilizes a mathematical model to simulate the behavior of slime molds in close proximity to food, and its position update model (*x_new_*) is shown below:(5)$$x_{new}=\left\{\begin{array}{c} rand\cdot (ub-lb),\\ x_{b}(t)+v_{b}(w\cdot xm(t)-xn(t)),\\ vc\cdot x(t), \end{array}\begin{array}{c} rand<z\\ r\leq p\\ r>p \end{array}\right.$$
(6)$$p=\tanh\left|s(i)-DF\right|,i=1,2,\cdots ,k$$
where *ub* and *lb* are the upper and lower bounds of the search space; *rand* is a random number in [0, 1]; *v_b_* is a random value in [−a, a]; *v_c_* is a random value that decreases linearly from 1 to 0; *t* is the number of the current iteration; *x_b_* is the coordinate with the highest concentration of the currently searched odors; *x* is the coordinate of the slime molds; *x_m_* and *x_n_* are the coordinates of two arbitrarily selected slime molds; *w* is the mass of the slime molds; *s*(*i*) is the fitness number of *x*; and *DF* is the optimal fitness number obtained in the optimization process.

The parameter *a* is mathematically modeled as:(7)$$a=\arctan h\left(-\frac{t}{T}+1\right)$$
where *T* is the maximum number of iterations.

The mathematical model of *w* is shown below:(8)$$w(\mathrm{smellindex}(i))=\left\{\begin{array}{c} 1+r\cdot \mathrm{lg}\left(\frac{bF-s(i)}{bF-wF}+1\right)\\ 1+r\cdot \mathrm{lg}\left(\frac{bF-s(i)}{bF-wF}+1\right) \end{array}\right.$$
smell index = *sort* (*s*)(9)
where *r* is a random number within [0, 1]; *bF* is the best fitness number in the current iteration; *wF* is the worst fitness number in the current iteration; and smell index is the fitness number.

### 2.6. SMA-LSSVM Prediction Model

Suykens [27] proffered least-squares support vector machine (LSSVM) as an improvement on support vector machine (SVM), which reduces the computational difficulty and effectively advances the training speed and accuracy.

The penalty parameter γ and the kernel parameter in the LSSVM prediction model are essential for the prediction accuracy. Thus, the screening of ideal eigenvalues in the effectiveness of the prediction model takes much importance [28]. In this paper, according to the competitiveness of the SMA in different search environments, two kernel features of the least-squares support vector machine are optimized to improve the prediction precision of the model.

### 2.7. Model Construction Method and Accuracy

Based on the magnitude of correlation coefficients, fractional-order differential spectra and fractional-order differential spectral indices suitable for evaluating SPAD were screened, and the LSSVM and SMA-LSSVM models were selected for estimating the wheat vertical distribution of SPAD. The optimal results from the model were selected for the current fertility period, which provides technical support for the detection of SPAD on a large scale under wheat-field-planting conditions. To estimate the performance of different models, 75% of the sample data were employed as the training set of the model and the remaining 25% of the sample data were applied as the validation set. Coefficient of determination (R^2^) and root mean square error (RMSE) were chosen to assess the performance and accuracy of the model and were calculated as follows.
(10)$$R^{2}=1-\sqrt{\frac{\sum_{i}{\left(y_{\mathrm{true}}-y_{\mathrm{pred}}\right)}^{2}}{\sum_{i}{\left(y_{\mathrm{true}}\right)}^{2}}}$$
(11)$$\mathrm{RMSE}=\sqrt{\frac{\sum_{i}{\left(y_{\mathrm{true}}-y_{\mathrm{pred}}\right)}^{2}}{m}}$$
where *y*_true_ and *y*_pred_ are the predicted and actual SPAD values, respectively; m is the number of samples.

## 3. Results and Discussion

### 3.1. Analysis of SPAD Content at Different Spatial Vertical Scales

In this experiment, three groups of data were collected simultaneously for SPAD_RL_, SPAD_EL_ and SPAD_CL_ at the booting stage, and the vertical distributions of the three types of wheat are shown below.

In Figure 4, the relative chlorophyll content SPAD_RL_ > SPAD_PL_ > SPAD_CL_ was observed at the booting stage, which indicates that the relative chlorophyll content was accumulated when the spatial vertical scale is deepened. This conclusion can be roughly drawn from the comparison of the maximum, minimum and mean in Table 1. Similarly, the standard deviation and variance of SPAD showed different patterns due to the varieties, with SPAD_PL_ < SPAD_RL_ < SPAD_CL_, SPAD_RL_ < SPAD_PL_ < SPAD_CL_ and SPAD_RL_ < SPAD_CL_ < SPAD_PL_ for Jinqiang 11, Jinqiang 9 and Longchun 23, respectively, suggesting that the relative chlorophyll contents of the three varieties accumulated when the spatial vertical scale is deepened. Spatial scale deepens, the SPAD of leaves increases, the dispersion of SPAD shows different patterns due to the three varieties of wheat and the relative chlorophyll content is not evenly distributed. According to Figure 4, due to the distribution of anomalies and the disparity from normal values of wheat, only SPAD_CL_ and its spectral data were used in this study.

### 3.2. Fractional-Order Differential Spectroscopy

#### 3.2.1. Fractional-Order Differential Spectral Downscaling

In this study, the 400–1000 nm spectral data of 48 samples (of which 2 samples were lost) were processed by 0–1-order fractional-order differentiation at an interval of 0.1, and 46 samples were utilized for the correlation analysis between fractional-order differentiated spectra and SPAD. Although the hyperspectral data reflect subtle differences in spectral features, the strong correlation between bands, especially between neighboring bands, leads to a great redundancy of information, making conventional analysis algorithms ineffective. Therefore, this study utilizes the standard normal variate transformation (SNV) [29] to preprocess the spectral reflectance, and then realizes the feature selection through the successive projections algorithm (SPA) [30]. As shown in Figure 5, in general, after SNV treatment of the original spectrum, the differences between the spectra are reduced, and they are more tight and consistent. The successive projection algorithm involves the utilization of the projection analysis of wavelength vectors and is combined with the correction model to select feature wavelengths, which can greatly reduce the pressure of model training. The optimal number of wavelengths was determined by the principle of RMSE minimum in original reflectance samples of wheat, and 33 spectral bands were finally screened out from 400–1000 nm for the subsequent model building.

#### 3.2.2. Correlation of Fractional-Order Differential Spectra with SPAD

The 33 screened spectral bands were subjected to Pearson correlation analysis with SPAD to obtain a table of correlations between canopy spectral reflectance and SPAD.

As can be seen from Table 2, 29 of the 33 bands screened by wheat spectral reflectance reached highly significant levels (*p* < 0.01) with the measured SPAD, but the correlation coefficients were distributed in the range of 0.419–0.475, which was generally low.

On the basis of analyzing the relationship between fractional-order differential spectra and SPAD_CL_ at the booting stage, the bands with the greatest correlation with SPAD_CL_ in each fractional-order spectrum were extracted (Table 3).

In Table 3, the differential processing of reflectance spectral data can improve the sensitivity of the spectra to SPAD_CL_, and the correlation coefficient of 401.53 nm in the original spectral reflectance with SPAD was the strongest, with a correlation coefficient of 0.475, and the correlations of 0.2, 0.4 and 0.9 orders of differential spectra with SPAD_CL_ were improved by 3.6%, 9.3% and 2%, respectively, compared to the original reflectance spectral data. In particular, the 0.4-order differential spectrum at 443.2 nm had the highest correlation with SPAD_CL_. Combining the analytical results in Table 1 and Table 2, the 0.4-order 443.2 nm (r = −0.568), 0.2-order 976.66 nm (r = 0.511), 0.9-order 415.05 nm (r = −0.495), 0.2-order 983.97 nm (r = −0.460), 0.1-order 718.28 nm (r = −0.462) and 0.2-order 718.85 nm (r = −0.440) were sequentially superimposed as inputs of the prediction model according to the superiority of the correlation in order to find the optimal differential spectra-based wheat SPAD detection model.

### 3.3. The Construction of Fractional-Order Differential Spectral Index

In order to determine the fractional order suitable for wheat relative chlorophyll and its corresponding bands, two-band fractional-order differential spectral indices sensitive to SPAD_CL_ were constructed, and the correlation between the fractional-order differential spectral indices and SPAD_CL_ was analyzed at the order of 0 to 1.0, respectively.

In Figure 6, when the order is 0.1–1.0, the combined regions of 850–990 nm wavelength Ri and 430–490 nm wavelength Rj, the combined regions of 850–990 nm wavelength Ri and 840–860 nm wavelength Rj are the sensitive regions for FDI to monitor the SPAD of wheat. The sensitive regions of FRI and FNDI to SPAD are larger than that of FDI, as the sensitive regions of FRI are focused on 440–480 nm wavelength Ri and 440–480 nm wavelength Ri and 440–650 nm wavelength Rj and 440–650 nm wavelength Ri. The sensitive areas of FNDI are located at 670–790 nm Ri and 640–760 nm Rj and 590–670 nm Ri and 460–500 nm Rj.

For the sake of screening the optimal orders of the three fractional-order differential spectral indices and their corresponding bands and to determine the specific expressions of the fractional-order differential spectral indices FDI, FRI and FNDI, the band combinations with the maximum correlation coefficients between the three fractional-order differential spectral indices and wheat SPAD at the orders from 0 to 1.0 were further extracted in this study (Table 4).

The correlation between the fractional-order differential spectral index FDI of order 0.2 to 1.0 and wheat SPAD is better than that of the original spectral index, with the optimal order of 0.7, corresponding to the band combinations of 989.59 and 488.35 nm. The correlation between the spectral index FRI of order 0.4 to 1.0 and wheat SPAD is better than that of the original spectral index, with the optimal order of 0.7 and corresponding band combinations of 475.12 and 841.72 nm. The correlation between the spectral index FNDI of 0.5 to 0.7 orders and wheat SPAD was superior to that of the original spectral index, and FNDI constructed with two bands of 0.6-order differential spectra of 468.81 and 851.65 nm was the best. As the order increases, the three fractional-order differential spectral indices gradually increase and then gradually decrease regarding the SPAD sensitivity. Meanwhile, it can be seen that among the three two-band fractional-order differential spectral indices FDI, FRI and FNDI constructed in this study, the correlation between FDI and FRI and SPAD of wheat was better than that of FNDI. The reason was that, at the booting stage, the chlorophyll content gradually increased, and it began to supply nutrients to the spikes, and FDI and FRI were more sensitive to the chlorophyll content.

The comprehensive analysis of Table 4 and Figure 6 shows that the effect of fractional-order differentiation on spectral processing is enhanced by the synergistic effect of two-dimensional bands, and the correlation between the fractional-order differentiated spectral indices constructed by different mathematical transformations and SPAD is enhanced compared to the single-band reflectance. Therefore, in this study, based on the three fractional-order differential indices constructed, FDI (989.59, 488.35) (r = 0.724), FSI (475.12, 841.72) (r = 0.700), FSI (475.12, 888.7) (r = 0.696), FDI (989.59. 483.74) (r = 0.691), FDI (989.59, 473.65) (r = 0.690), FDI (639.87, 619.73) (r = 0.690), FDI (989.59, 488.35) (r = 0.688), FSI (899.77, 468.81) (r = −0.687) and FDI (989.59, 474.07) (r = 0.686) were sequentially employed as inputs for the predictive model to find the optimal wheat SPAD detection model based on differential spectral indices.

### 3.4. Model Construction and Evaluation

#### 3.4.1. Fractional-Order Differential Spectral Reflectance

The differential spectra with strong correlation in Section 3.2 were ordered from highest to lowest and gradually used as input variables, and finally 3~5 differential spectra were screened out to establish LSSVM and SMA-LSSVM prediction models with wheat SPAD to construct the wheat SPAD estimation model and evaluate its accuracy. The wheat SPAD prediction models established based on 3~5 differential spectra are shown in Table 5.

Comparing the constructed LSSVM and SMA-LSSVM prediction models, the SPAD prediction model constructed by taking 0.4-order 443.2 nm, 0.2-order 976.66 nm, 0.9-order 415.05 nm, 0.2-order 983.97 nm and 0.1-order 718.28 nm as the input variables has the highest accuracy, with R^2^ = 0.924 and RMSE = 2.735, so the SMA-LSSVM prediction model with these five differential spectra as input variables better reflects the ability of the fractional-order differential spectra combination to predict SPAD.

#### 3.4.2. Fractional-Order Differential Spectral Index

The differential spectral indices with strong correlation in Section 3.3 were ordered from highest to lowest and gradually used as input variables, and finally 7~9 differential spectral indices were screened out to establish LSSVM and SMA-LSSVM prediction models with wheat SPAD to construct the wheat SPAD estimation model and to evaluate its accuracy. The wheat SPAD prediction models established based on 7~9 differential spectral indices are shown in Table 6.

Comparing the constructed LSSVM and SMA-LSSVM prediction models, the SPAD prediction model constructed by using the eight differential spectral indices as input variables has the highest accuracy, with R^2^ = 0.997 and RMSE = 0.093, so the SMA-LSSVM prediction model under these eight differential spectral indices as input variables better reflects the ability of the fusion of differential spectral indices in predicting the SPAD.

#### 3.4.3. Feature Fusion

In this study, in view of the fractional-order differential spectral index and fractional-order differential spectra screened in Section 3.4.1 and Section 3.4.2 as the inputs to the model, the wheat SPAD detection model is constructed using the LSSVM and SMA-LSSVM algorithms, and the results are shown in Table 7.

Table 7 shows the results of SPAD evaluation by fusing spectra and spectral indices. Compared with the fractional-order differential spectral index (R^2^ = 0.997, RMSE = 0.093) (Table 6), the fractional-order differential spectra did not evaluate SPAD better, with R^2^ and RMSE reaching up to 0.924 and 2.735 (Table 5). Further fusion of the fractional-order differential spectral index and band spectra to build a model for wheat SPAD assessment improved over fractional-order differential spectroscopy but was not better than the fractional-order differential spectral index; therefore, comparison of the two validations yielded that the fractional-order differential spectral index was more suitable than fractional-order differential spectroscopy for assessing wheat SPAD and that fusion of the two did not improve the capability to assess SPAD in wheat.

In an effort to further evaluate the model precision and generalization ability, the constructed models were analyzed in this study using the validation set, from which it can be seen that the difference in the prediction precision of the wheat SPAD estimation models constructed employing the preferred fractional-order differential indices was smaller in the validation set compared to the test set, with the R^2^ between the predicted and measured SPAD of the SMA-LSSVM model constructed by fractional-order differential spectral indices of 0.931 and RMSE of 2.754, which was 23.1% higher than that of the LSSVM model and the RMSE was 0.854 lower, which indicated that the predictability of this model was very good and the results were reliable. The adaptive competence of the model is very strong, which can improve the response of spectra to SPAD and the capability of information mining, strengthen the correlation between SPAD and spectra and better characterize the wheat growth information. The reason why the improved LSSVM model based on the slime mold algorithm has the highest accuracy is that the SMA-LSSVM algorithm can better balance the fitting ability and generalization ability of the model by optimizing the objective function.

### 3.5. Discussion

Through the above analysis, it was found that the correlation between the original spectral data processed by fractional-order differentiation and the measured wheat SPAD was improved, and the correlations between the 0.2-, 0.4- and 0.9-order differential spectra and the SPAD_CL_ were improved by 3.6%, 9.3% and 2%, respectively, compared with that of the original reflectance spectral data, and in particular, the correlation between the 0.4-order differential spectra at 443.2 nm and SPAD_CL_ was the highest, reaching −0.568. Although the response of the 0.1~1-order spectral reflectance to the measured SPAD was not obvious, it reflected that the reflectance spectra were differentially processed to improve the sensitivity to a certain extent and that the differential processing of the hyperspectral data at fractional order was able to highlight the fine information of the spectra, describe nuance among the spectral data, enhance the absorption characteristics of the spectra to some extent and retain more effective information. Owing to the number and quality of the samples, the conclusion has limitations. This paper is based on SNV preprocessing and SPA, so different kinds of preprocessing and dimensionality reduction methods are to be investigated in the future.

In view of the original empirical spectral index and fractional-order differential processing, the optimal fractional order and wavelength reflecting the SPAD of wheat were determined and improved, and the two-waveband fractional-order differential index was constructed. Through the above experiments, it was found that the correlation between the fractional-order differential spectral index (FDI) and wheat SPAD at the order of 0.1~1.0 was better than that of the original spectral index, and the optimal order was 0.7, corresponding to the wavelength combinations of 989.59 and 488.35 nm, with a correlation coefficient of 0.724. Compared with the 0.4-order differential spectra at 443.2 nm wavelength, the correlation coefficient was improved by 27.5%, which indicates that the fractional-order differential spectral index is better than the spectral index and that it is more suitable to reflect the SPAD of wheat. It indicates that the processing role of fractional-order differentiation on spectra is enhanced by the synergistic effect of two-dimensional bands, and the correlation between fractional-order differentiated spectral indices constructed by different mathematical transformations and SPAD is improved compared with single-band reflectance.

In this study, the two-band differential spectral index and the optimal fractional-order wavelength were used as input variables of the LSSVM prediction model, which were input sequentially according to the maximum correlation, and the number of filtered features was judged according to the precision of the prediction model and the two were mixed as the feature factor to establish the SPAD prediction model for wheat. Finally, we screened the SMA-LSSVM model with the highest accuracy under the combination of 0.4-order 443.2 nm, 0.2-order 976.66 nm, 0.9-order 415.05 nm, 0.2-order 983.97 nm and 0.1-order 718.28 nm with accuracy of the test and validation sets of 0.924 and 0.915, respectively. Whereas for the two-band differential spectral indices, we screened eight characteristic factors in the LSSVM prediction model and found that it accurately realizes the wheat SPAD evaluation (R^2^ = 0.997, RMSE = 0.093; R^2^ = 0.931, RMSE = 2.754). The optimal fractional-order wavelengths are fused, and the results of the SPAD evaluation based on the LSSVM are not improved, which suggests that the fusion of the two-band differential spectral index and optimal fractional-order wavelength is not an efficacious method to promote the SPAD detection accuracy, and due to the quality of the samples, the degree of response of the optimal fractional-order wavelengths to the measured indices partially influences the fusion effect.

This method is also applied to other crops such as corn, tomatoes, etc. For example, Ding et al. [31] established multiple linear regression models using sensitive spectral bands of tomato chlorophyll content under four different preprocessing methods. The model accuracy ranged from high to low, including envelope removal, absorbance, raw and first-order differentiation. Han [32] indicated that compared with the original spectrum, the results of the LNC model using fractional-order differential preprocessing are closer, and the correlation coefficients of the optimal band combination of the two are greater than 0.8. However, the fractional-order differential spectrum has a significant performance improvement compared to the original spectrum in the PNC model. However, this paper further refines integer-order differentiation and applies fractional-order differential spectral indices, resulting in higher prediction accuracy of the model compared to previous studies.

In summary, the correlation between spectral reflectance and SPAD is improved by fractional-order differentiation of the original spectra, and the three kinds of two-band fractional-order differential spectral indices constructed are able to sensitively reflect the measured SPAD, which are improved compared with the optimal fractional-order wavelengths. Among the SMA-LSSVM models according to two-band differential spectral indices, the optimal fractional-order wavelengths and a mixture of the two as input variables, the SMA-LSSVM prediction model on account of the two-band differential spectral index is the best, with R^2^ and RMSE of 0.997, 0.093, 0.931 and 2.754 for the test and validation sets, respectively.

## 4. Conclusions

In this study, on the basis of fractional-order differentiation of canopy reflectance spectra, we determined the optimal fractional-order and its wavelength that can sensitively reflect SPAD, constructed a two-band fractional-order differentiated spectral index and employed them as input variables for the LSSSVM and SMA-LSSVM models to establish a wheat SPAD prediction model.

The correlation between spectral reflectance and wheat SPAD was promoted by fractional-order differentiation of the raw spectra, and the maximum correlation coefficient was of the order of 0.4, corresponding to a wavelength of 433.2 nm, with a value of −0.568, which is 9.3% higher than the maximum correlation coefficient of the original spectra.Among the three fractional-order differential spectral indices, the optimal correlation between the single-band and wheat SPAD was −0.568, and the optimal correlation coefficients between the FDI, FRI and FNDI and wheat SPAD were 0.724, 0.700 and 0.680, respectively, suggesting that the two-band differential spectral indices can reflect wheat SPAD more sensitively than the optimal fractional-order wavelengths.The SMA-LSSSVM prediction model constructed on the basis of two-band fractional-order differential spectral indices is superior to the optimal fractional-order wavelength and the mixture of the two and better realizes the accomplishment of wheat SPAD.

## Figures and Tables

**Figure 1 sensors-24-01693-f001:**
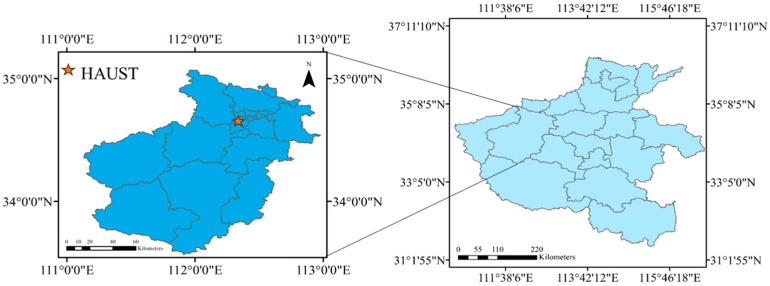
Experimental station of Henan University of Science and Technology.

**Figure 2 sensors-24-01693-f002:**
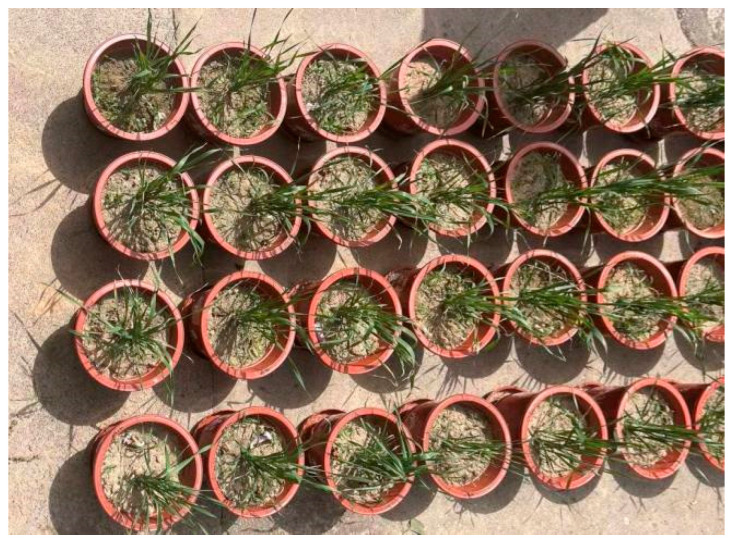
Experimental site.

**Figure 3 sensors-24-01693-f003:**
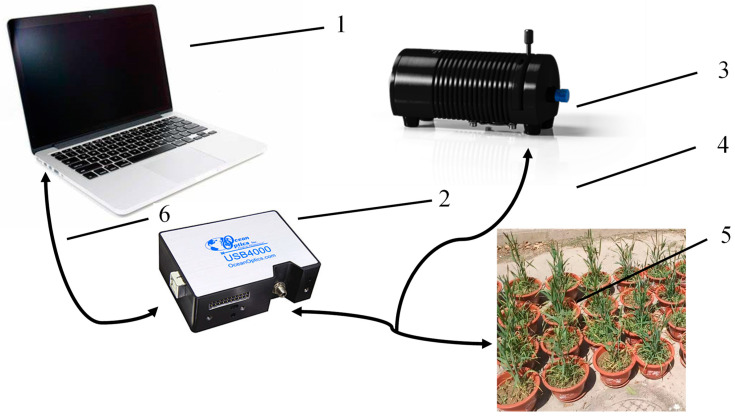
The general architecture of the wheat-phenotyping system. 1. a computer. 2. an Ocean Insight spectrometer. 3. a light source. 4. a reflection probe. 5. wheat. 6. a cable.

**Figure 4 sensors-24-01693-f004:**
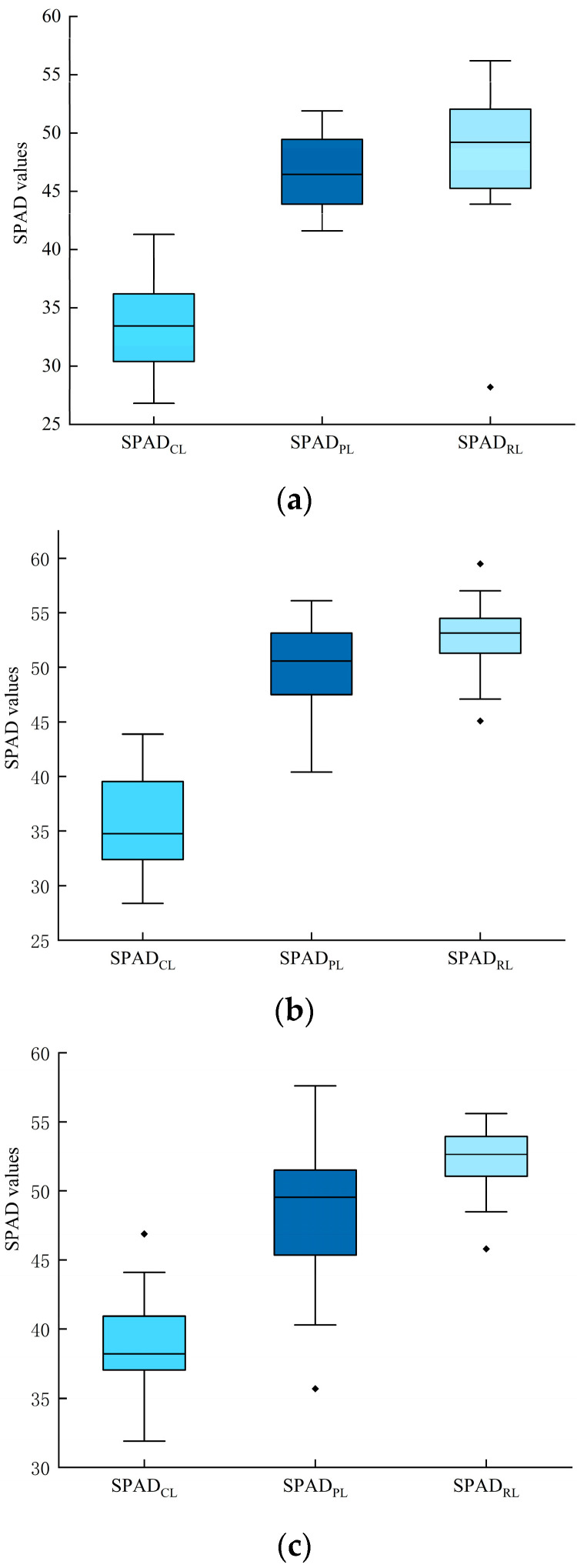
Wheat SPAD observations of wheat on booting stage. The middle line of the box indicates the median of the data, the upper and lower lines of the box indicate the upper and lower quartiles and the upper and lower whiskers of the box indicate the maximum and minimum of the data, respectively. (**a**) The SPAD observations of Jinqiang 11. (**b**) The SPAD observations of Jinqiang 9. (**c**) The SPAD observations of Longchun 23.

**Figure 5 sensors-24-01693-f005:**
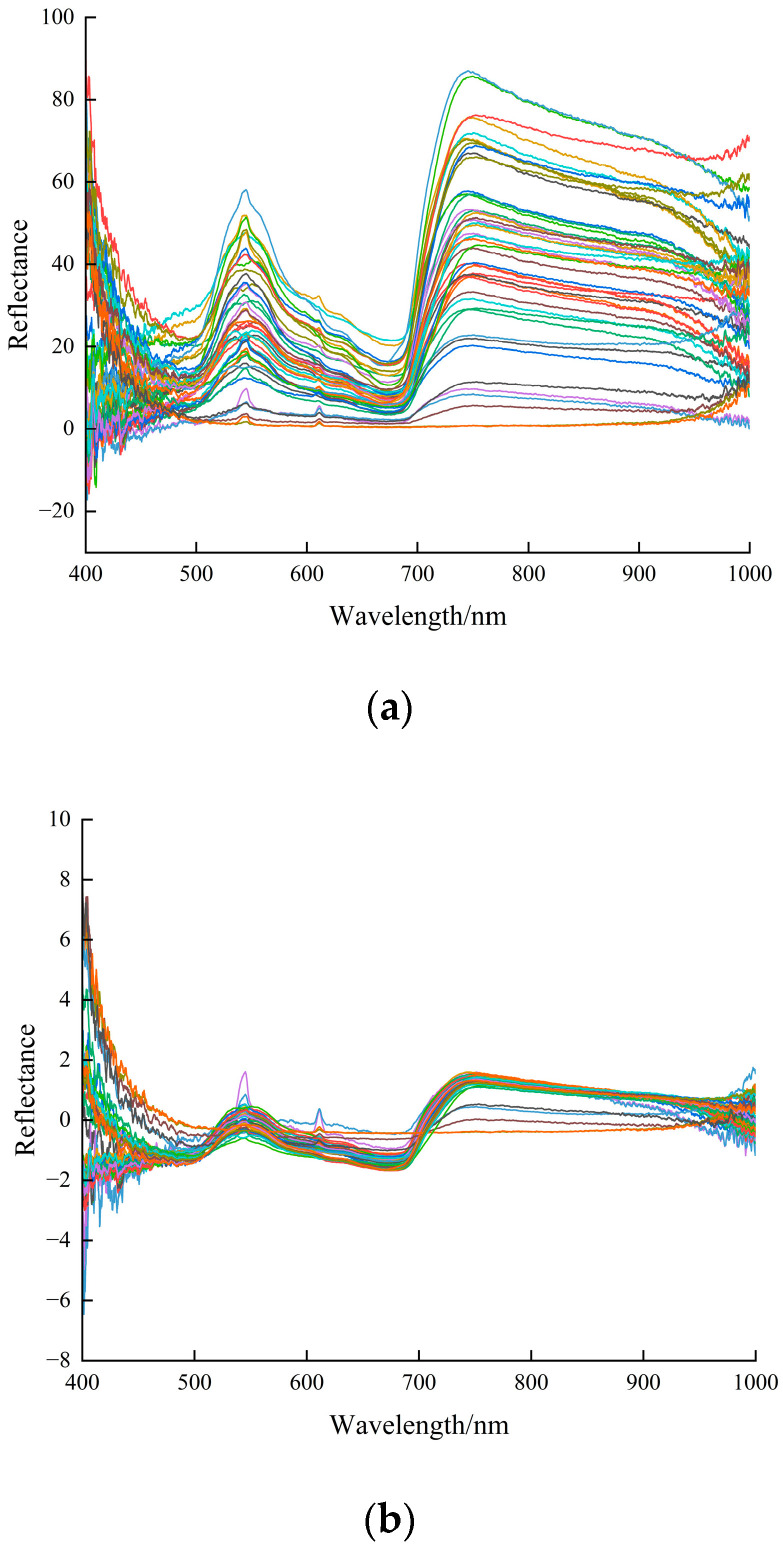
Bands screened with SPA. (**a**) Original spectrum. (**b**) SNV preprocessing results. (**c**) Variables selected.

**Figure 6 sensors-24-01693-f006:**
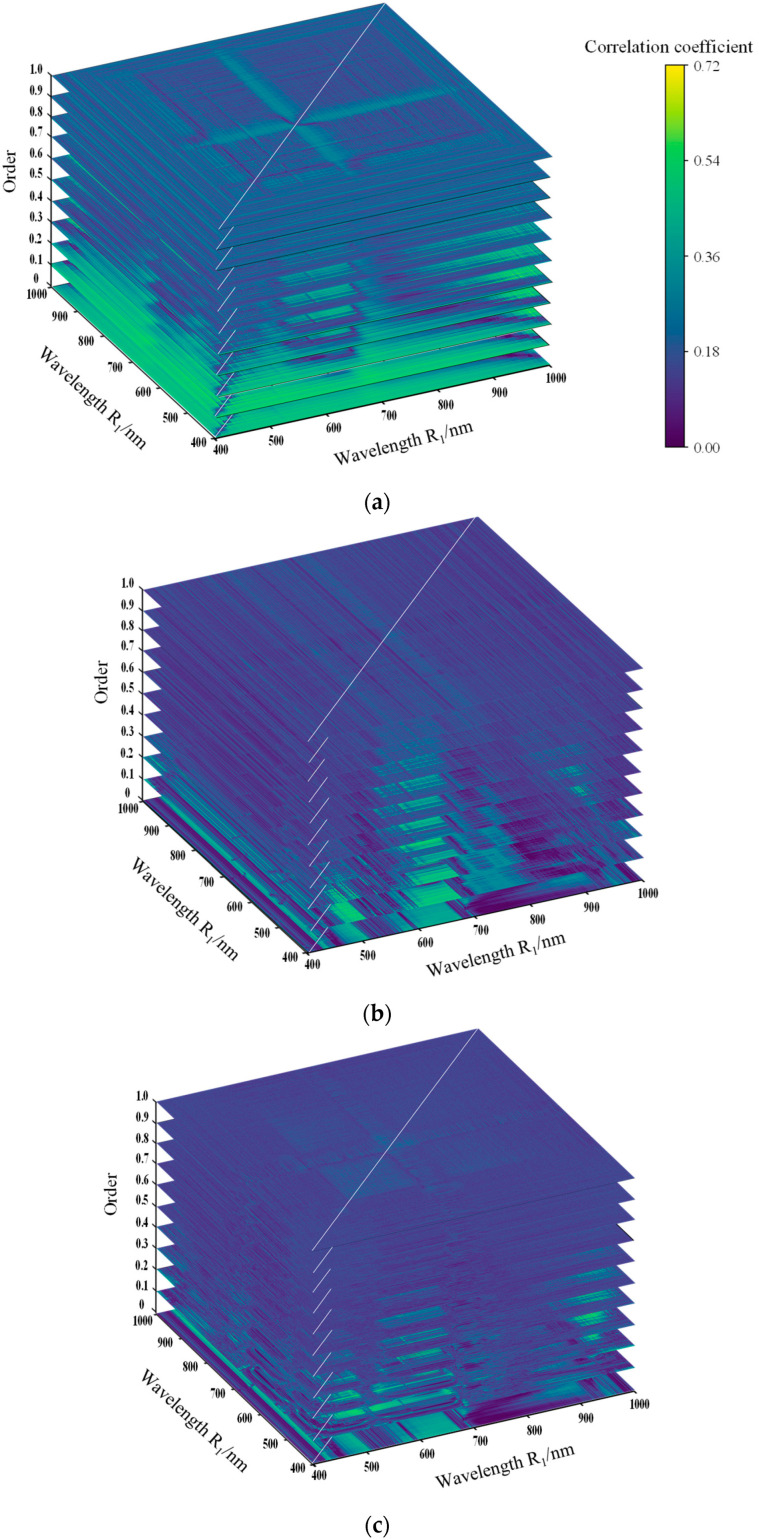
Correlation between fractional differential spectral indices and SPAD. (**a**) FDI. (**b**) FRI. (**c**) FNDI.

**Table 1 sensors-24-01693-t001:** SPAD data statistics.

DS	Maximum	Minimum	Mean	Standard Deviation
SPAD_CL(11)_	41.3	26.8	33.57	4.29
SPAD_PL(11)_	51.9	41.6	46.63	3.24
SPAD_RL(11)_	56.2	43.9	49.29	3.75
SPAD_CL(9)_	43.9	28.4	35.75	4.57
SPAD_PL(9)_	56.1	40.4	50.16	3.92
SPAD_RL(9)_	59.5	45.1	52.68	3.53
SPAD_CL(23)_	46.9	31.9	38.76	3.79
SPAD_PL(23)_	57.6	35.7	48.43	5.27
SPAD_RL(23)_	55.6	45.8	52.18	2.68

**Table 2 sensors-24-01693-t002:** Correlation coefficients between the original spectral screening bands and measured SPAD.

Wavelength/nm	Cc	Wavelength/nm	Cc	Wavelength/nm	Cc	Wavelength/nm	Cc
408.4	0.453 **	410.76	0.441 **	407.76	0.442 **	401.53	0.475 **
408.19	0.447 **	410.12	0.444 **	709.22	−0.452 **	434.05	0.419 **
414.62	0.444 **	411.62	0.422 **	424.88	0.420 **	412.69	0.420 **
414.4	0.449 **	409.69	0.444 **	414.83	0.443 **	491.07	0.101
412.47	0.431 **	411.19	0.452 **	542.97	−0.312 *	993.05	0.419 **
415.26	0.425 **	413.97	0.441 **	991.07	0.430 **	970.98	0.302 *
403.03	0.457 **	402.39	0.467 **	1000.11	0.448 **		
415.47	0.430 **	403.46	0.453 **	408.61	0.447 **		
401.74	0.470 **	409.04	0.446 **	690.42	−0.056		

Note: Cc is correlation coefficient. ** indicates a significant correlation at the 0.01 level; * indicates a significant correlation at the 0.05 level. (the same below).

**Table 3 sensors-24-01693-t003:** Maximum correlation between fractional-order differential spectra and SPAD and their corresponding wavelengths.

Orders	Correlation Coefficient	Wavelength/nm	Orders	Correlation Coefficient	Wavelength/nm
0.1	0.462	718.28	0.6	0.375	441.71
0.2	0.511	976.66	0.7	0.432	442.35
0.3	0.423	400.45	0.8	0.405	455.5
0.4	−0.568	443.2	0.9	0.495	415.05
0.5	0.414	442.78	1.0	0.422	747.7

**Table 4 sensors-24-01693-t004:** Correlation coefficients of the optimal fractional-order differential spectral indices with SPAD for the two bands and the corresponding band combinations.

Orders	FDI		FRI		FNDI	
	Correlation Coefficient	Band Combination /nm	Correlation Coefficient	Band Combination /nm	Correlation Coefficient	Band Combination /nm
0	0.618	(509.86, 510.07)	0.645	(838.46, 822.98)	0.644	(838.46, 822.98)
0.1	0.614	(858.12, 848.58)	0.590	(864.03, 850.21)	0.590	(864.03, 850.21)
0.2	0.634	(858.12, 849.67)	0.571	(861.52, 845.7)	0.590	(516.72, 966.46)
0.3	0.674	(989.59, 438.1)	0.610	(443.63, 450.2)	0.632	(444.47, 937.33)
0.4	0.662	(968.98, 442.78)	0.676	(443.41, 912.68)	0.637	(468.81, 937.33)
0.5	0.685	(927.72, 468.81)	0.696	(475.12, 888.7)	0.672	(468.81, 927.72)
0.6	0.691	(989.59, 483.74)	−0.687	(899.77, 468.81)	0.680	(468.81, 851.65)
0.7	0.724	(989.59, 488.35)	0.700	(475.12, 841.72)	0.672	(468.81, 824.62)
0.8	0.669	(989.59, 488.35)	−0.671	(870.05, 628.52)	0.637	(675.38, 754.14)
0.9	0.661	(837.37, 853.63)	0.650	(996.34, 411.62)	0.622	(789.04, 649.19)
1.0	0.667	(989.59, 437.89)	0.649	(475.12, 643.25)	0.619	(453.59, 751.87)

**Table 5 sensors-24-01693-t005:** The wheat SPAD prediction models established based on 3~5 differential spectra.

Differential Spectral Indices	Algorithm	Test		Validation	
		R^2^	RMSE	R^2^	RMSE
3 differential spectra	LSSVM	0.914	3.119	0.888	4.494
	SMA-LSSVM	0.914	3.124	0.889	4.421
4 differential spectra	LSSVM	0.910	3.250	0.889	4.439
	SMA-LSSVM	0.924	2.735	0.915	3.411
5 differential spectra	LSSVM	0.909	3.282	0.894	4.255
	SMA-LSSVM	0.900	3.625	0.906	3.744

**Table 6 sensors-24-01693-t006:** The wheat SPAD prediction models established based on 7~9 differential spectral indices.

Differential Spectral Indices	Algorithm	Test		Validation	
		R^2^	RMSE	R^2^	RMSE
7 differential spectral indices	LSSVM	0.934	2.405	0.909	3.643
	SMA-LSSVM	0.989	0.409	0.937	2.504
8 differential spectral indices	LSSVM	0.939	2.202	0.910	3.611
	SMA-LSSVM	0.997	0.093	0.931	2.754
9 differential spectral indices	LSSVM	0.936	2.303	0.909	3.646
	SMA-LSSVM	0.959	1.474	0.927	2.906

**Table 7 sensors-24-01693-t007:** Fusion spectra and spectral indices.

Algorithm	Test		Validation	
	R^2^	RMSE	R^2^	RMSE
LSSVM	0.942	2.093	0.896	4.175
SMA-LSSVM	0.981	0.684	0.903	3.899

## Data Availability

The data presented in this study are available on request.

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
