# Peer review of "Research on SPAD Estimation Model for Spring Wheat Booting Stage Based on Hyperspectral Analysis"

_sensors, 2024, doi:10.3390/s24051693_

Round 1

Reviewer 1 Report

Comments and Suggestions for Authors

The paper is generally an interesting study, but there are some problems as follows:

1, the wheat in this paper was only sown in March, which does not seem to be in line with the laws of wheat cultivation in Henan Province, and secondly, are the authors sure that wheat sown in March enters the spike stage in April.

2, the text needs to add whether there is irrigation treatment.

3, the text of the icon format needs to be adjusted, some did not align!

4, chapter 3.1 part of the significance of the analysis, than the standard deviation and variance can better illustrate the problem.

5, the conclusion of the part is not rigorous enough, the paper is only for the spad of wheat spad research, why not for the reproductive period between the greening and maturity of the comprehensive analysis, expand the depth of the study, so that it is more persuasive.

6, the discussion is not deep enough, the paper only analyses the phenomenon in the text, while the discussion focuses on the same points of the study and other studies, as a way to show the significance of the article research.

Comments on the Quality of English Language

Suggested refinement of language

Reviewer 2 Report

Comments and Suggestions for Authors

In this study, on the basis of fractional order differentiation of canopy reflectance spectra, the authors determined the optimal fractional order and its wavelength that can sensitively reflect SPAD, constructed a two-band fractional order differentiated spectral index, and employed them as an input variable for the LSSSVM and SMA-LSSVM models to establish a wheat SPAD prediction model. The comments for the improvement of the manuscript are as follows: 

1.What are the advantages of applying least squares support vector machine to predict SPAD and improving the least squares support vector machine with slime mold algorithm in this paper? Why not apply the least squares support vector machine directly. Clarify. 

2.Why is the nitrogen gradient set in the pot experiment like this? Please clarify it. 

3.The application of fractional order differentiation reflected in this paper has indeed improved the correlation between the spectrum and the measured parameters. What are the effects of this method on SPAD detection and other research on other crops? Please supplement it in the discussion. 

4.There are a few format issues with this article, such as formula 12 being actually formula 10 Please check the whole article and revise.

Reviewer 3 Report

Comments and Suggestions for Authors

1. It is necessary to present the general architecture of the proposed plant phenotyping system using hyperspectral images.

2. It is necessary to provide the technical characteristics of the instruments and equipment used to conduct these studies.

3. It would be advisable to present a calibration graph for chlorophyll, by which the accuracy of concentration measurements was assessed.

4. On the graphical dependencies presented in Figure 5, you must indicate the names of the axes and units of measurement.

Round 2

Reviewer 1 Report

Comments and Suggestions for Authors

The description of watering in the amended version is not appropriate, the unit of watering is generally mm, and what is the source of 300ml of watering each time, every 3 days should be watered once, which is unreasonable, and it is recommended to check it carefully.
